# The Prevalence and the Impact of Frailty in Hepato-Biliary Pancreatic Cancers: A Systematic Review and Meta-Analysis

**DOI:** 10.3390/jcm11041116

**Published:** 2022-02-20

**Authors:** Klara Komici, Micaela Cappuccio, Andrea Scacchi, Roberto Vaschetti, Giuseppe Delli Carpini, Vito Picerno, Pasquale Avella, Maria Chiara Brunese, Giuseppe Rengo, Germano Guerra, Leonardo Bencivenga

**Affiliations:** 1Department of Medicine and Health Sciences, University of Molise, 86100 Campobasso, Italy; micaelacappuccio24@gmail.com (M.C.); scacchiandrea@me.com (A.S.); vaschetti.r@gmail.com (R.V.); giuseppedc16@gmail.com (G.D.C.); vito.picerno.97@gmail.com (V.P.); mariachiarabrunese@libero.it (M.C.B.); germano.guerra@unimol.it (G.G.); 2Department of Translational Medical Sciences, University of Naples “Federico II”, 80131 Naples, Italy; giuseppe.rengo@unina.it; 3Istituti Clinici Scientifici Maugeri SpA Società Benefit (ICS Maugeri SpA SB), 82037 Telese Terme, Italy; 4Department of Advanced Biomedical Sciences, University of Naples “Federico II”, 80131 Naples, Italy; leonardobencivenga@gmail.com; 5Gérontopôle de Toulouse, Institut du Vieillissement, CHU de Toulouse, 31300 Toulouse, France

**Keywords:** frailty, elderly, mortality, hepatic cancer, biliary cancer, pancreatic cancer

## Abstract

Background: Frailty has been associated with increased mortality among hepatobiliary pancreatic (HBP) cancer patients. Nevertheless, estimates of frailty prevalence in HBP cancers and the precise average effect regarding mortality remains uncertain. The present systematic review and meta-analysis aimed to quantify: (1) the prevalence of frailty in patients with liver and pancreatic cancers and (2) the impact of frailty on mortality in patients affected by liver and pancreatic cancers. Methods: MEDLINE/PubMed database search was conducted from inception until 1 November 2021, the pooled prevalence and relative risk (RR) estimate were calculated. Results: A total of 34,276 patients were identified and the weighted prevalence of frailty was 39%; (95% [C.I.] 23–56; I2 = 99.9%, *p* < 0.0001). Frailty was significantly associated with increased mortality RR 1.98 (95% [C.I.] 1.49–2.63; I2 = 75.9%, *p* = 0.006). Conclusions: Frailty prevalence is common among HBP cancer patients and exerts a significant negative impact on survival. These findings are characterized by significant heterogeneity and caution is warranted on their interpretation. However, stratification of patients with HBP cancer by frailty status may provide prognostic information and may inform priorities for decision-making strategy.

## 1. Introduction

Frailty is a clinical syndrome characterized by multiple reductions in physiological reserves and vulnerability to stressors [1]. New drugs, infections, surgery, and hospitalizations are common stressors which may trigger significant changes in health status of frail patients, leading to several negative health outcomes and mortality.

Even though a standardized frailty measurement tool is still lacking, different frailty models have been developed, with the Fried phenotype model and cumulative deficit model, based on comprehensive geriatric assessments (CGA), being the most common [2,3]. Frailty is associated with advanced age, physical and cognitive decline, multiple chronic conditions and polypharmacotherapy.

Oncologic patients are characterized by several features of frailty syndrome, such as reduction in physiological reserves, low physical performance, malnutrition, cachexia. Moreover, chemotherapy and surgical treatment may induce adverse outcomes [4]. Almost one-third of cancers are diagnosed in patients aged 70 years or over, making the therapeutic approach and the prognosis in this population particularly challenging [5].

According to recent epidemiological data, liver and pancreatic cancers counted, respectively, 905, 677, and 495,773 new cancer cases in 2020 worldwide, and estimation of cancer death was 830,180 for liver and 466,003 for pancreas malignances [6]. Recent data have shown that surgery procedures for the elderly with hepatic and liver cancers may be well tolerated, but frailty concurs to influence the decision-making strategies [7,8]. A recent consensus statement suggested that frailty may be associated with negative short- and long-term outcomes in patients with hepatobiliary and pancreatic malignancies [9]. Frailty assessment in this population may contribute to disease management; accordingly, detection of other systems and organ deficits before treatment strategy reduces the risk of complications. Indeed, CGA has shown a predictive role in treatment toxicity and complications. However, epidemiological data regarding the prevalence and the impact of frailty in this population are lacking. Therefore, in this study, we aimed to quantitatively synthesize, through a systematic review and meta-analysis, (1) the prevalence of frailty in patients with liver and pancreatic cancers and (2) the impact of frailty on mortality in patients affected by liver and pancreatic cancers.

## 2. Materials and Methods

This systematic review and meta-analysis were performed according to the Preferred Reporting Items for Statistic Reviews and Meta-Analyses (PRISMA) statement and Strengthening the Reporting of Observational Studies in Epidemiology (STROBE) [10,11]. This study followed a pre-determined unpublished protocol available upon request.

### 2.1. Search Strategy

Studies were identified and evaluated independently by two authors in the MEDLINE/PubMed database, until 1 November 2021. Free text terms and or MeSH terms were used as keywords for the search strategy referred to frailty, geriatric assessments, pancreatic cancer, biliary cancer, and liver cancer. In detail, we used the combination of the following search terms: geriatric assessment AND (biliary cancer OR liver cancer OR pancreatic cancer) and frailty AND (biliary cancer OR liver cancer OR pancreatic cancer). Each of these combinations provided the following complete search items. The complete search items which result from the above-mentioned combinations are: ((“geriatric assessment”[MeSH Terms] OR (“geriatric”[All Fields] AND “assessment”[All Fields]) OR “geriatric assessment”[All Fields]) AND (“biliary tract neoplasms”[MeSH Terms] OR (“biliary”[All Fields] AND “tract”[All Fields] AND “neoplasms”[All Fields]) OR “biliary tract neoplasms”[All Fields] OR (“biliary”[All Fields] AND “cancer”[All Fields]) OR “biliary cancer”[All Fields] OR (“liver neoplasms”[MeSH Terms] OR (“liver”[All Fields] AND “neoplasms”[All Fields]) OR “liver neoplasms”[All Fields] OR (“liver”[All Fields] AND “cancer”[All Fields]) OR “liver cancer”[All Fields]) OR (“pancreatic neoplasms”[MeSH Terms] OR (“pancreatic”[All Fields] AND “neoplasms”[All Fields]) OR “pancreatic neoplasms”[All Fields] OR (“pancreatic”[All Fields] AND “cancer”[All Fields]) OR “pancreatic cancer”[All Fields]))) AND (1000/1/1:2021/11/1[pdat]); and ((“frailty”[MeSH Terms] OR “frailty”[All Fields] OR “frailties”[All Fields]) AND (“biliary tract neoplasms”[MeSH Terms] OR (“biliary”[All Fields] AND “tract”[All Fields] AND “neoplasms”[All Fields]) OR “biliary tract neoplasms”[All Fields] OR (“biliary”[All Fields] AND “cancer”[All Fields]) OR “biliary cancer”[All Fields] OR (“liver neoplasms”[MeSH Terms] OR (“liver”[All Fields] AND “neoplasms”[All Fields]) OR “liver neoplasms”[All Fields] OR (“liver”[All Fields] AND “cancer”[All Fields]) OR “liver cancer”[All Fields]) OR (“pancreatic neoplasms”[MeSH Terms] OR (“pancreatic”[All Fields] AND “neoplasms”[All Fields]) OR “pancreatic neoplasms”[All Fields] OR (“pancreatic”[All Fields] AND “cancer”[All Fields]) OR “pancreatic cancer”[All Fields])) AND 1000/01/01:2021/11/01[Date—Publication]) AND (1000/1/1:2021/11/1[pdat]).

### 2.2. Selection Criteria

All selected titles and abstracts were reviewed by two authors independently. Studies were considered eligible if they fulfilled the following criteria: aim 1: (a) they reported the prevalence of frailty in patients with pancreatic cancer; (b) they reported the prevalence of frailty in patients with liver or biliary cancer; aim 2: (a) studies which reported relevant analysis regarding liver and or pancreatic cancer mortality in patients with and without frailty. Only articles published in English language were considered. Articles reporting data on pancreatic and periampullary cancers were also considered for this study. Exclusion criteria were (a) abstracts, editorials, comments, unpublished data; (b) unclear frailty definition; (c) frailty assessment only based on single measure such as gait speed, grip strength, or muscle mass; and (d) studies where data regarding the number of death events were not stratified on frailty status, and where calculation of relative risk (RR) was not possible.

### 2.3. Quality and Risk of Bias Assessment

Quality of the included studies was assessed using the guidelines in the Newcastle–Ottawa Scale (NOS) [12], based on selection, comparability, exposure, or endpoint. These items were categorized into three major components containing eight items. Presence of publication bias was explored visually performing the test for asymmetry of the funnel plot by Egger test [13].

### 2.4. Data Extraction

Two reviewers, independently using a standardized form, completed data extraction. Disagreement was resolved by consensus and by the opinion of a third reviewer when necessary. Information on study year, author first name, data regarding sample size, prevalence, setting, outcome, and characteristics of pancreatic or liver cancers was recorded.

### 2.5. Statistical Analysis

The prevalence of frailty was summarized using descriptive statistics. Pooled prevalence rates accounting for inter-study variation were analyzed using a non-linear random effects model and statistical uncertainties were expressed in 95% Confidence Intervals (CI). RR estimates together with CI were extracted from each study and a pooled overall average effect size was calculated using random effect models. Heterogeneity was assessed using I2 statistic. Heterogeneity has been considered substantial if I2 value was greater than 25% [14]. To explore the reasons for heterogeneity subgroup analysis was conducted for: (a) studies identifying frailty with Fried Frailty Phenotype; (b) studies identifying frailty with Modified Frailty Index; (c) data regarding pancreas cancer; and (d) data regarding liver cancer. In addition, to explore the influence of potential effect modifiers on endpoints, a meta-regression analysis was performed to test age and sex (male%). For all meta-regression analyses, a random effects model was used to take into account the mean of a distribution of effects across studies. All reported test results were two-tailed and a *p* value ≤ 0.05 was considered significant. Data analyses were performed with STATA version 16 (StataCorp LLC, College Station, TX, USA).

## 3. Results

A total of 310 articles were identified by the initial search (Figure 1), 43 manuscripts were retrieved for more detailed evaluation, and 18 studies [7,15,16,17,18,19,20,21,22,23,24,25,26,27,28,29,30] were finally included in the systematic review qualitative and quantitative analysis. Relevant data regarding mortality were reported in four studies. Detailed characteristics of the included studies are reported in Table 1.

### 3.1. Prevalence of Frailty in Patients with Pancreatic and Liver Cancer

A total of 34,276 patients were identified with an average age ranging from 62 to 85.3 years and a rate of male population ranging from 37.9% to 76%. Characteristics of studies included in the meta-analysis are reported in Table 1. The weighted prevalence of frailty in patients with pancreas and liver cancer was 39%, (95% [C.I.] 23–56; I2 = 99.9%, *p* < 0.0001) (Figure 2). Meta-regression analysis revealed that age (beta coefficient 0.004 95% CI −0.0202–0.030 *p* = 0.697) and male population: beta coefficient −0.006 (95% [C.I.] −0.022–0.009 *p* = 0.408) were not significant moderators.

### 3.2. Subgroup Analysis

Estimated prevalence of frailty in patients with pancreas cancer was 42% (95% [C.I.] 19–64; I2 = 100%, *p* < 0.0001) (Appendix A), and in patients with liver cancer was 29% (95% [C.I.] 11–48; I2 = 92%, *p* < 0.0001) (Appendix A).

The Fried Frailty Model was used for the detection of frailty in three studies and the overall prevalence was 41% (95% [C.I.] 18–64; I2 = 90.1%, *p* < 0.0001) (Figure 3). The Modified Frailty Index was performed in six studies and estimated prevalence of frailty across these studies was 53% (95% [C.I.] 24–82; I2 = 100%, *p* < 0.0001) (Figure 4).

### 3.3. Frailty Is Associated with Increased Mortality in Patients with HPB Cancer

On the basis of data from four studies, frailty was significantly associated with increased mortality RR 1.98 (95% [C.I.] 1.49–2.63; I2 = 75.9%, *p* = 0.006) (Figure 5). In studies where frailty detection was performed by the Modified Frailty Index, effect estimate was increased by 25.8%: RR 2.49 (95% [C.I.] 2.02–3.07; I2 = 0%, *p* = 0.663) (Figure 6). Subgroup analysis of studies where only surgery treatment was performed also revealed that frail HBP patients, compared to non-frail HPB patients, are characterized by increased mortality: RR: 1.79 [C.I.] 1.35–2.39; I2 = 67.4%, *p* = 0.046 (Appendix A).

### 3.4. Study Quality

The quality of the included studies evaluated by NOS criteria was moderate or good, ranging from 5 to 8 points. NOS quality assessment of the included studies are reported in Appendix A.

### 3.5. Publication Bias

Asymmetry was assessed by visual inspection of funnel plots. However, Egger’s regression test did not indicate significant publication bias among the included studies. For aim 1, overall prevalence *p* = 0.291; subgroup analysis frailty prevalence in pancreas cancer *p* = 0.421; liver cancer *p* = 0.735; and aim 2, *p* = 0.334 (funnel plots in Appendix A Appendix A).

## 4. Discussion

The present systematic review and meta-analysis summarize for the first time the prevalence of frailty among patients affected by HPB cancers. The evaluation of data regarding 34,276 patients revealed that frailty prevalence accounts to about 39% in this population. Furthermore, frailty exerts an adverse role in overall mortality, as demonstrated by RR 1.98 95% C.I. 1.49–2.63. However, the interpretation of our findings is limited by the different definitions and criteria used to identify frailty.

Frailty is a wide-range metric of general health status and multiple physiological reserves, which strongly correlates with patient prognosis [31,32]. The prevalence of frailty in community dwelling population ranges from 4 to 59%, based on the criteria used for the definition and identification of frailty [33]. Frailty evaluation in single diseases is also reported to range widely, for instance from 9–28% in chronic obstructive pulmonary disease, 52.2–85.9% in hypertension, and 15–52% in heart failure [34,35]. Furthermore, frailty prevalence in patients with advanced liver disease awaiting liver transplantation ranges from 17–43% [36]. 

In our study, the overall estimates of frailty prevalence were 39% (95% [C.I.] 23–56%) considering all patients affected by pancreas and liver cancers. A previous meta-analysis study focused on frailty prevalence in all oncologic patients revealed that the median prevalence of frailty is about 42% ranging from 6 to 86% [4]. Our finding regarding overall prevalence is in line with this study, even if most of the articles were on breast, prostate, and colon-rectal cancers [4]. Considering that breast and prostate cancers are characterized by high incidence in ageing population, it is intuitive to expect a high prevalence of frailty in these populations. Pancreas and liver cancer are not as frequent as breast, prostate, or colon-rectal cancers world-wide; nevertheless, the high prevalence of frailty that we find in this study may be explained by the increased incidence, proportional to chronological age. Indeed, cholangiocarcinoma’s incidence increases with age, peaking at 59–75 years for males and 80–84 years for females [37], while median age for hepatocellular carcinoma onset in Europe, Japan, and North America is over 62 years [38]. In most of the evidence, the incidence of pancreatic cancer increases with age, occurring mostly after the age of 70 and only 32% of patients are diagnosed under 64 years in the U.S.A. [39]. However, in our findings, definition and screening of frailty was performed by a huge variety of assessment scales, such as the Modified Frailty Index, Fried Frailty index, short physical performance battery, clinical frailty scale, G8 score, and VES-13 scale. It should be mentioned that not all the above-mentioned assessment tools focus on the same aspects of frailty and heterogeneity which have been detected in the methods used to identify frailty. Indeed, Kojima et al, reported that quantification of frailty is performed by dissimilar tools even in the same clinical setting and, surprisingly, application of the same method still produced a wide range of frailty prevalence [40]. Of note, a study focused on the prevalence and feasibility of different frailty screening tools in nursing homes concluded that conceptualization of frailty leads to a significant heterogeneity in the prevalence of frailty, which significantly affects the interrelation between multimorbidity and disability [41]. Despite this, some studies report that different frailty tools have shown a similar capacity to detect frailty and similar prognostic potentialities [42].

Pathophysiology of frailty is characterized by reduced reserves of different inter-related systems and organs such as: brain, endocrine, immune, musculoskeletal, cardiovascular, respiratory, and renal. Frailty syndrome is often characterized by extreme fatigue, un-explained weight loss, and fluctuating disability [1]. Delirium and falls are further consequences, commonly associated with hospitalization which leads to the development of severely impaired mobility [43,44]. Significant weight loss and fatigue are frequent symptoms in oncologic patient. Moreover, pain, dysphagia, and reduced absorption of nutrients, which characterize clinical presentation of pancreas and liver cancer [45], may lead to malnutrition [46] and the development of muscle mass impairment. Typical biological substrate of physical frailty is represented by fatigue, malnutrition, reduced muscle mass, impairment of physical performance, and mobility. Limited functional reserve, impairment of liver function, and chemotherapy may contribute to the development of muscle mass reduction and sarcopenia [47,48,49,50]. 

In addition, chronic inflammation and immune system modulation are closely related to liver and pancreatic cancers progression [51,52].

Frail HBP cancer patients, compared to non-frail HPB cancer patients, suffer from an increased risk of mortality, revealing that frailty is a significant predictor of mortality in this population. Accumulation of health deficits related to functional status, mobility, malnutrition, and comorbidities are all elements of frailty, apart from the chronological age, which may influence the overall survival [53]. Poor physical performance is strongly associated with disability and adverse surgical outcomes [54,55]. A meta-analysis study revealed that sarcopenia was associated with an increased risk of complications after gastrointestinal tumor resection and suggested that combination of physical performance and muscle mass measurements may increase the prognostic value and accuracy in preoperative risk stratification [47].

Sub-group analysis of studies whose frailty assessment was based on the Modified Frailty Index, which takes into account comorbidities and functional status, increased the effect estimates: RR 2.49 (95% [C.I.] 2.07–3.11). Furthermore, analysis of data in HPB patients who had undergone surgery also revealed that frailty is significantly associated with increased mortality: RR: 1.79 (95% [C.I.] 1.35–2.39).

The advances made in medical care have provided an important rise of elderly population needing oncologic and surgical management. Despite substantial improvements in perioperative management and surgical techniques, pancreatic and hepatobiliary surgery carries a high risk of morbidity and mortality [56,57]. As matter of the fact, frailty screening it is not routinely performed among HBP cancer patients, however findings from our meta-analysis strongly encourage clinicians to perform CGA in this population, as it represents the only method to evaluate the complexity which characterizes elderly patients with malignancies. Furthermore, frailty evaluations may provide relevant information about multiple accumulated deficits as malnutrition, physical mobility impairment and reduced cognitive performance; furthermore, it may also exert preventive role, as pre-rehabilitation procedures and new developing strategies may be applied in selected patients [58,59,60,61,62]. The oncological outcomes of laparoscopic liver resection in elderly patients with colorectal metastasis were comparable to those observed with open resection, and a reduction in both minor and major postoperative morbidity was observed [63,64]. 

In addition, some perioperative parameters such as drain removal and refeeding seem to favor robotic pancreaticoduodenectomy compared to open surgery in frail patients [30] and, in selective cases, robotic surgery may be considered for colorectal cancer liver metastases [65,66,67].

### Limitations

Our data are characterized by increased heterogeneity, as demonstrated by I2 evaluation of above 45%. It should be mentioned that different frailty detection tools used in different studies may, in part, explain the high heterogeneity. In addition, our analysis is based on observational studies, which may be characterized by increased heterogeneity. Another limitation of our results is that data from the included studies were not sufficient to provide subgroup analysis based on pancreas or liver cancer subtypes. Furthermore, it was not possible to compare different treatment strategies.

## 5. Conclusions

The present systematic review and meta-analysis study summarizes the prevalence and the effects of frailty on overall mortality among patients with HBP cancers. Frailty prevalence is high and exerts a negative role on survival of HBP cancer patients. These findings are characterized by significant heterogeneity, and lack of a standard definition of frailty hampers their interpretation. However, stratification of patients with HBP cancer based on comprehensive geriatric assessment tools may provide prognostic information and may critically contribute to decision-making strategy.

## Figures and Tables

**Figure 1 jcm-11-01116-f001:**
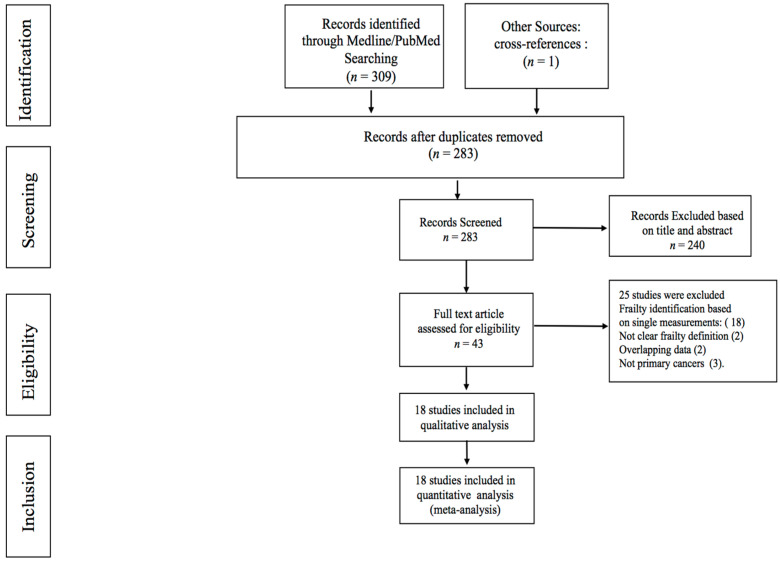
Flowchart of search strategy and included studies.

**Figure 2 jcm-11-01116-f002:**
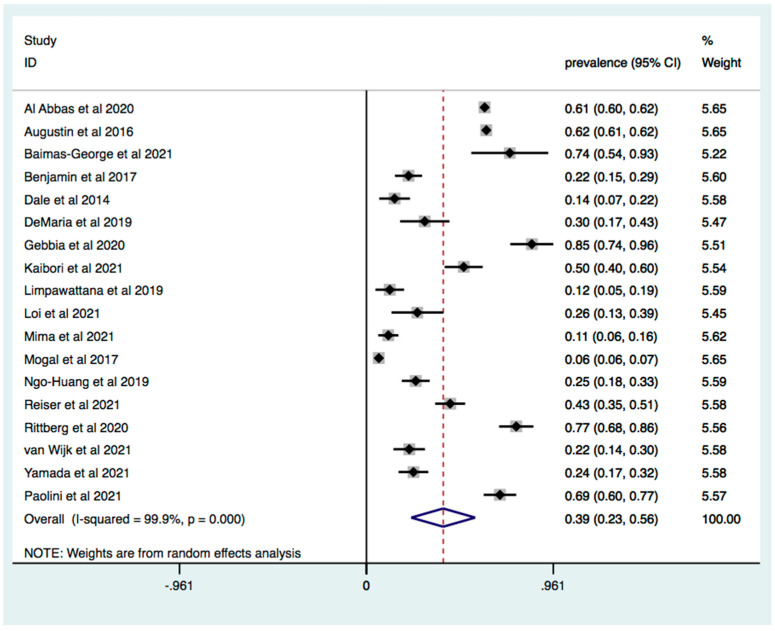
Overall prevalence of frailty in patients with HBP cancer. Forest plot of cumulative prevalence of frailty from all included studies. Squares are study-specific prevalence. Diamond is the pooled prevalence. Horizontal lines represent 95% Confidence Interval (CI).

**Figure 3 jcm-11-01116-f003:**
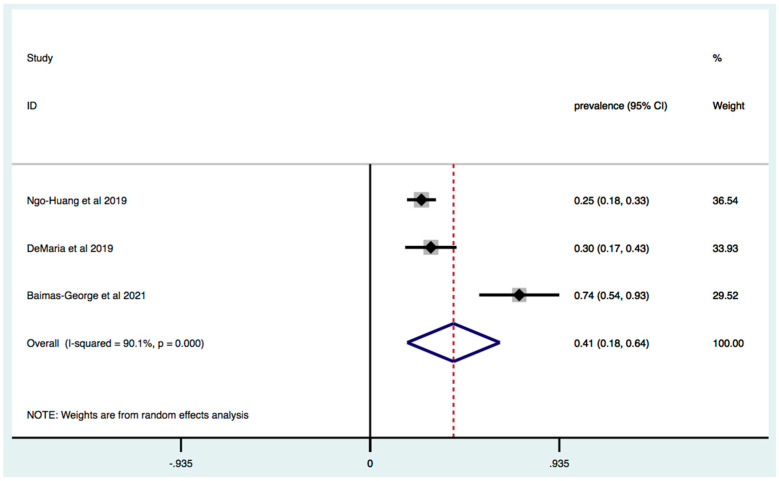
Prevalence of frailty in patients with HPB cancer: identification of frailty based on Fried Frailty Criteria. Forest plot of cumulative prevalence of frailty from all studies applying Fried Phenotype Model. Squares are study-specific prevalence. Diamond is the pooled prevalence. Horizontal lines represent 95% Confidence Interval (CI).

**Figure 4 jcm-11-01116-f004:**
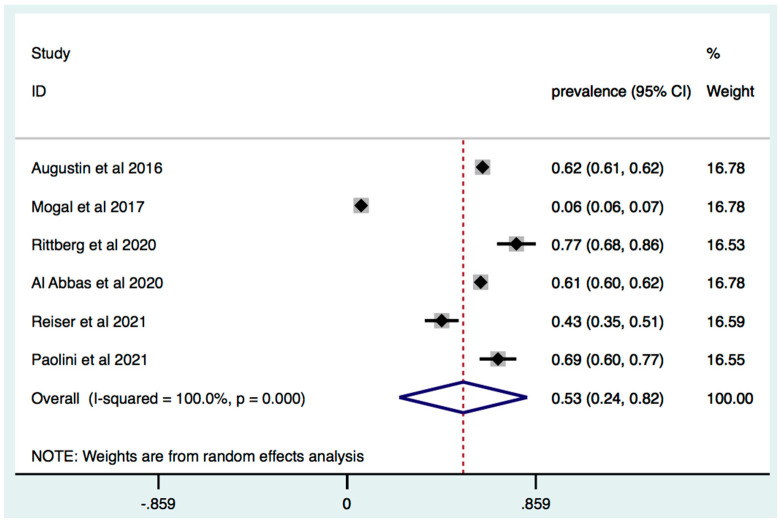
Prevalence of frailty in patients with HPB cancer: identification of frailty based on Modified Frailty Index. Forest plot of cumulative prevalence of frailty from all studies applying Modified Frailty Index. Squares are study-specific prevalence. Diamond is the pooled prevalence. Horizontal lines represent 95% Confidence Interval (CI).

**Figure 5 jcm-11-01116-f005:**
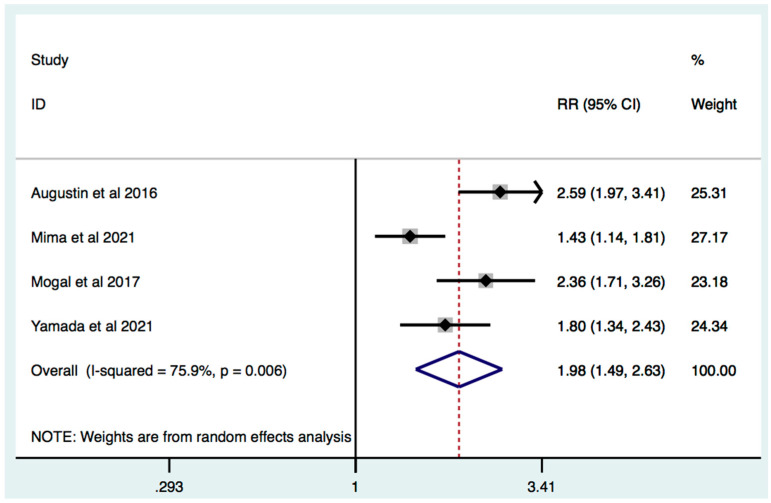
Effect of frailty on mortality in patients with HPB cancer. Forest plot of mortality risk comparing frailty hepatobiliary pancreatic cancers vs. non-frail hepatobiliary pancreatic cancer patients. Squares are study-specific Relative Risk (RR). Diamond is the estimated overall RR. Horizontal lines represent 95% Confidence Interval (CI).

**Figure 6 jcm-11-01116-f006:**
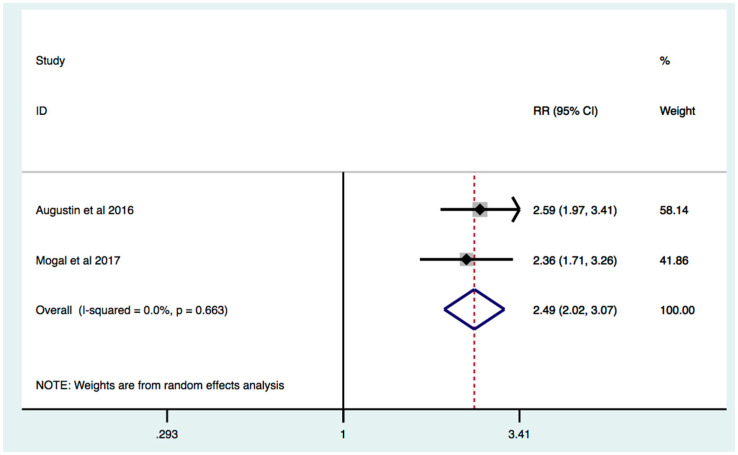
Effect of frailty on mortality in patients with HPB cancer: identification of frailty based on Modified Frailty Index. Forest plot of mortality risk comparing frailty hepatobiliary pancreatic cancers vs. non-frail hepatobiliary pancreatic cancer patients when frailty was identified by Modified Frailty Index. Squares are study-specific Relative Risk (RR). Diamond is the estimated overall RR. Horizontal lines represent 95% Confidence Interval (CI).

**Table 1 jcm-11-01116-t001:** Characteristics of included studies.

First Author and Year	Study Design	Total Population	Mean Age	Male (%)	Frail	Non-Frail	Frailty Tool	Cancer Type	Treatment
Al Abbas et al., 2020	Retrospective Cohort Study	9867	64.5	53.2	5996	3871	Modified Frailty Index	Pancreatic Cancers (Adenocarcinoma 50%)	Surgery, Chemotherapy, Radiotherapy
Augustin et al., 2016	Retrospective Cohort Study	13,020	N/R	48.3	8024	4996	Modified Frailty Index	Pancreatic Cancer, not specified	Surgery, Chemotherapy, Radiation
Baimas-George et al., 2021	Prospective Cohort Study	19	62	47	14	5	Fried Phenotype Model	Pancreatic Adenocarcinoma (90%); Colangiocarcinoma (10%)	Chemotherapy, Surgery
Benjamin et al., 2017	Prospective Cohort Study	134	65.4	52	29	105	SPPB	Pancreatic Ductal Adenocarcinoma	Surgery
Dale et al., 2014	Prospective Cohort Study	76	67.3	56.3	11	65	VES-13, Fried and SPPB	Pancreatic Endocrine, Exocrine, Biliary	Surgery
DeMaria et al., 2019	Prospective Cohort Study	50	64	68	15	35	Fried Phenotype Model	Hepatocellular Carcinoma	Surgery, liver transplantation
Gebbia et al., 2020	Prospective Cohort Study	40	74.7	65	34	6	G8	Advanced/metastatic Pancreatic Carcinoma	Chemotherapy
Kaibori et al., 2021	Retrospective Cohort Study	100	79	N/R	50	50	G8, VES-13	Hepatocellular Carcinoma	Surgery
Limpawattana et al., 2019	Retrospective Cohort Study	75	N/R	77.3	9	66	Frail Scale	Biliary Cancer	Chemotherapy
Loi et al., 2021	Retrospective Cohort Study	42	85.3	N/R	11	31	G8	Hepatocellular Carcinoma	SBRT
Mima et al., 2021	Retrospective Cohort Study	142		56	16	126	Clinical Frailty Scale	Pancreatic Cancer: Adenocarcinoma (98%)	Surgery
Mogal et al., 2017	Retrospective Cohort Study	9986	64.1	51.2	637	9349	Modified Frailty Index	Pancreatic Cancer, not specified	Surgery
Ngo-Huang et al., 2019	Prospective Cohort Study	142	65	65.5	36	106	Fried Phenotype Model	Pancreatic Ductal Adenocarcinoma	Surgery, Chemotherapy, Radiation, Palliative
Reiser et al., 2021	Retrospective Cohort Study	158	N/R	37	68	90	Modified Frailty Index	Pancreatic Ductal Adenocarcinoma	Surgery, Neoadjuvant therapy
Rittberg et al., 2020	Retrospective Cohort Study	87	73.7	54	67	20	Modified Frailty Index	Advanced Pancreatic Cancer	Chemotherapy
van Wijk et al., 2021	Prospective Cohort Study	100	74	51	22	78	Groningen frailty indicator	Hepatobiliary pancreatic cancers (Mixed population)	Scheduled for surgery
Yamada et al., 2021	Retrospective Cohort Study	120	N/R	N/R	29	91	Clinical Frailty Scale	Pancreatic Ductal Adenocarcinoma	Surgery
Paolini et al., 2021	Retrospective Cohort Study	118		52.5	81	37	Modified Frailty Index	Pancreatic, periampullary cancers, common bile duct cancers	Surgery (open or robotic)

## Data Availability

Data supporting the results of the study will be available on request.

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
