# Peer review of "The Prevalence and the Impact of Frailty in Hepato-Biliary Pancreatic Cancers: A Systematic Review and Meta-Analysis"

_jcm, 2022, doi:10.3390/jcm11041116_

Round 1

Reviewer 1 Report

The authors performed a systematic review and meta-analysis of studies reporting on frailty in patients with HPB cancers and its impact on mortality. This is an important clinical topic. The statistical analysis is well done, however I am concerned about the decision to pool all the heterogeneous data and synthesize it using meta-analysis (and meta-regression).

Major comments:

  • The statistical analysis is well done, however that does not fix the major problem with this study, which is clinical heterogeneity (especially with regard to frailty definitions and the patient populations). While we have tools to adjust for statistical heterogeneity, there is no way to adjust for clinical heterogeneity. As such, I would be very cautious about synthesizing these results using meta-analysis. If you decide to proceed with the meta-analysis, I would highlight the limitations in the interpretation very heavily (including in abstract, conclusion, and discussion).
  • For a systematic review, you need to clearly show your systematic search process. Kindly present all the search term combinations that were used, so that someone can reproduce the exact same results you present in your flow diagram (figure 1).
  • I recommend that the discussion is restructured and enriched. Most journals recommend 5-6 separate paragraphs discussing separate ideas/topics.

Other comments:

  • Page 2 lines 53-55: Where do these numbers apply? In Europe? Worldwide? Please clarify.
  • There are a few typos throughout the manuscript.

Author Response

February 13th 2022

Reviewer 1

The authors performed a systematic review and meta-analysis of studies reporting on frailty in patients with HPB cancers and its impact on mortality. This is an important clinical topic. The statistical analysis is well done, however I am concerned about the decision to pool all the heterogeneous data and synthesize it using meta-analysis (and meta-regression).

Major comments:

The statistical analysis is well done, however that does not fix the major problem with this study, which is clinical heterogeneity (especially with regard to frailty definitions and the patient populations). While we have tools to adjust for statistical heterogeneity, there is no way to adjust for clinical heterogeneity. As such, I would be very cautious about synthesizing these results using meta-analysis. If you decide to proceed with the meta-analysis, I would highlight the limitations in the interpretation very heavily (including in abstract, conclusion, and discussion).

Reply: Thank You for the comments. The Reviewer's concern regarding the heterogeneity is important. Even though many validated tools for Frailty have been identified, a standard diagnostic tool for frailty is lacking. Application of different frailty screening tools, which evaluate different characteristics and domains of frailty may produce important heterogeneity. Despite this, it has been reported that different frailty tools have shown similar capacity to detect frailty and similar prognostic potentialities Woo J et al. Comparison of frailty indicators based on clinical phenotype and the multiple deficit approach in predicting mortality and physical limitation. J Am Geriatr Soc. 2012 Aug;60(8):1478-86. doi: 10.1111/j.1532-5415.2012.04074.x.Following the reviewer suggestion, we highlighted this limitation regarding interpretation of the results in abstract, discussion and conclusions. Please find the above modifications in the text: 

Abstract line 32-34:  ''These findings are characterized by significant heterogeneity and caution is warranted on their interpretation.''

Discussion

line 454-455:  ''However, the interpretation of our findings is limited by the different definitions and criteria used to identify frailty.''

line 435-448 : ''However, in our findings definition and screening of frailty was performed by numerous assessment scales like modified frailty index, Fried frailty index, short physical per-formance battery, clinical frailty scale, G8 score and VES-13 scale. It should be mentioned that not all the above-mentioned assessment tools focus on the same aspects of frailty and heterogeneity has been detected in the methods used to identify frailty. Indeed, Kojima at al.,[40] reported that quantification of frailty is performed by dissimilar tools even in the same clinical setting and surprisingly, application of the same method still produced a wide range of frailty prevalence .Of note, a study focused on the prevalence and feasibility of different frailty screening tools in nursing homes, concluded that conceptualization of frailty leads to a significant heterogeneity in the prevalence of frailty which significantly affects the interrelation between multimorbidity and disability [41]. Despite this, some studies report that different frailty tools have shown similar capacity to detect frailty and similar prognostic potentialities [42].

Conclusions line 1031-1032:  ''These findings are characterized by significant heterogeneity and lack of a standard definition of frailty hampers their interpretation''.

For a systematic review, you need to clearly show your systematic search process. Kindly present all the search term combinations that were used, so that someone can reproduce the exact same results you present in your flow diagram (figure 1).

Reply:Following the reviewer suggestion, we added in the text the exact combination of the terms used for our search strategy which can be reproduced by the readers. Our analysis included articles present in Medline/PubMed till November 1ST, 2021. From the overall search result now 310 articles. Since we performed our research during November probably inadvertently we missed some articles which were not present during our search. However, we re-run the analysis and excluding duplicates 283 were screened. From this, in the final analysis we included also the article by Paolini et al  Robotic versus open pancreaticoduodenectomy: Is there any difference for frail patients? Surg Oncol. 2021 Jun;37:101515. doi: 10.1016/j.suronc.2020.12.009., which was missed in the previous analysis, and consequently we corrected tables and figures as appropriate.

I recommend that the discussion is restructured and enriched. Most journals recommend 5-6 separate paragraphs discussing separate ideas/topics.

Reply: Thank you for the comment. Discussion was restructured and enriched. In different paragraphs are discussed the following main points:

-frailty definition and prevalence and why frailty may be common in patients with HPB cancers  -- heterogeneity of our results;

-frailty pathophysiology and the links between frailty pathophysiology and HBP cancers  

-frailty impact in mortality and why frailty detection may be useful in patients with HPB.    

Other comments:

Page 2 lines 53-55: Where do these numbers apply? In Europe? Worldwide? Please clarify.

There are a few typos throughout the manuscript.

Reply: Worldwide epidemiological data regarding incidence and mortality in patients with cancers (reference nr.6).

We wish to thank the Reviewer for His/Her constructive criticism, suggestions and comments which helped us to significantly improve the quality of our study.

Reviewer 2 Report

  1. Please delete one word "immune" because it is repeated written in one sentence. (line 223-225)
  2. Since the authors selected 17 articles for analysis, the heterogeneity of the tool for evaluation of frailty should be mentioned in the discussion session.

Author Response

Reviewer 2

Please delete one word "immune" because it is repeated written in one sentence. (line 223-225)

Since the authors selected 17 articles for analysis, the heterogeneity of the tool for evaluation of frailty should be mentioned in the discussion session.

Reply:Thank you for reviewing our manuscript. The repeated word in line 223-225 was deleted.

In the revised version of our manuscript we discussed the heterogeneity in discussion section. Please check lines 435-448 : ''However, in our findings definition and screening of frailty was performed by numerous assessment scales like modified frailty index, Fried frailty index, short physical per-formance battery, clinical frailty scale, G8 score and VES-13 scale. It should be mentioned that not all the above-mentioned assessment tools focus on the same aspects of frailty and heterogeneity has been detected in the methods used to identify frailty. Indeed, Kojima at al.,[40] reported that quantification of frailty is performed by dissimilar tools even in the same clinical setting and surprisingly, application of the same method still produced a wide range of frailty prevalence .Of note, a study focused on the prevalence and feasibility of different frailty screening tools in nursing homes, concluded that conceptualization of frailty leads to a significant heterogeneity in the prevalence of frailty which significantly affects the interrelation between multimorbidity and disability [41]. Despite this, some studies report that different frailty tools have shown similar capacity to detect frailty and similar prognostic potentialities [42].

We wish to thank the Reviewer for His/Her suggestions and comments which helped us to significantly improve the quality of our study.

Reviewer 3 Report

Spelling and grammatical mistakes throughout document, would benefit from proof-reading through a native English speaker

Author Response

Reviewer 3

Spelling and grammatical mistakes throughout document, would benefit from proof-reading through a native English speaker

Reply:Thank you for reviewing our study and the good feedback. Grammatical mistakes, typos and English language was corrected in the revised version of our manuscript. 

We wish to thank the Reviewer for His/Her suggestions and comments which helped us to significantly improve the quality of our study.